# Relative Dose Intensity of Daratumumab, Lenalidomide, and Dexamethasone in Multiple Myeloma

**DOI:** 10.3390/cancers17030470

**Published:** 2025-01-30

**Authors:** Kazuhito Suzuki, Tadahiro Gunji, Masaharu Kawashima, Hideki Uryu, Riku Nagao, Takeshi Saito, Kaichi Nishiwaki, Shingo Yano

**Affiliations:** 1Division of Clinical Oncology/Hematology, Department of Internal Medicine, The Jikei University School of Medicine, Minato City 105-8461, Japan; 2Division of Clinical Oncology/Hematology, Department of Internal Medicine, The Jikei University Kashiwa Hospital, Kashiwa 277-8567, Japan

**Keywords:** multiple myeloma, daratumumab, lenalidomide, dexamethasone, relative dose intensity, survival, infection

## Abstract

Daratumumab (DARA), lenalidomide (LEN), and dexamethasone (DEX, DRd) are one of the standards of care for patients with multiple myeloma; however, the clinical impact of relative dose intensity (RDI) remains unclear. In this retrospective study, it was shown, using real-world data, that the median RDIs for LEN and DEX were low, whereas those for DARA were maintained. A high DARA RDI was associated with a long time to next treatment and overall survival (OS). A low RDI for DEX was also associated with long OS. The high RDI of DARA and low RDI of DEX reduced the incidence of severe infections. Multivariate analysis revealed that the RDIs of DARA and DEX were associated with a long survival time, independent of disease status and treatment response. In conclusion, the high RDI of DARA and low RDI of DEX predicted good clinical outcomes in patients with myeloma treated with DRd.

## 1. Introduction

Multiple myeloma (MM) is a heterogeneous group of plasma cell neoplasms that exhibit variability in morphology, phenotype, molecular biology, and clinical behavior. The development of novel agents such as proteasome inhibitors (PIs) and immunomodulatory drugs (IMiDs) has improved patient prognosis over the last decade; however, MM remains incurable [1]. Daratumumab (DARA), an anti-CD38 monoclonal antibody, exhibits conventional cytotoxicity, such as antibody- and complement-dependent cellular cytotoxicity and immunomodulatory activity [2,3]. DARA improves the clinical outcomes in newly diagnosed MM (NDMM) and relapsed and/or refractory MM (RRMM) [4,5,6,7,8,9]; thus, DARA-containing treatments are recommended according to several clinical guidelines. In the MAIA and POLLUX trials, overall survival (OS) and progression-free survival (PFS) in the DARA, lenalidomide (LEN), and dexamethasone (DEX, DRd) groups were significantly longer than those in the LEN and DEX (Rd) groups. The relative dose intensity (RDI) of anticancer therapeutics is generally associated with the clinical outcomes of aggressive hematological malignancies [10]. DEX contributes to anti-MM activity and immunosuppression, leading to an increased infection incidence and reduced immunological activity against MM cells [11]. Additionally, the RDI of the agents may be reduced in clinical practice compared to that in clinical trials [12,13]. Therefore, owing to their adverse effects, the doses of DEX and LEN should be reduced in clinical practice compared to those of DARA. However, the clinical significance of the RDIs of DARA, LEN, and DEX remains unclear.

In this retrospective study, the aim was to evaluate the clinical impact of the RDIs of DARA, LEN, and DEX in patients with NDMM and RRMM treated with DRd.

## 2. Materials and Methods

The medical records of patients with myeloma who underwent DRd at Jikei University Hospital and Jikei University Kashiwa Hospital between January 2018 and December 2022 and were followed-up until December 2023 were reviewed. This study was approved by the Independent Ethics Committee and Institutional Review Board of (34–404).

### 2.1. Patients

Patients aged ≥18 years who were diagnosed with symptomatic MM and received DRd participated in this study. In contrast, patients who underwent upfront autologous stem cell transplantation (ASCT) soon after DRd, had monoclonal gammopathy of undetermined significance, smoldering MM, isolated plasmacytoma, or primary plasma cell leukemia were excluded from this study. High-risk cytogenetic abnormality (HRCA) was defined as t(4;14), t(14;16), del17p, or 1q21 gain by fluorescence in situ hybridization [14].

### 2.2. Response Assessment

Disease response was assessed according to the International Myeloma Working Group criteria [15]. Minimal residual disease (MRD) status was analyzed using multicolor flow cytometry (SRL Inc., Tokyo, Japan) in patients whose monoclonal protein bands were not observed by two consecutive immunofixation electrophoreses (IFE) [16]. The cutoff value for MRD negativity was 1 × 10^−5^. The MRD assessment was repeated annually when the IFE status remained negative.

### 2.3. Dose Adjustment and RDI

The RDI was calculated by dividing the total dose by the estimated dose according to the MAIA and POLLUX trials [4,6]. Therefore, planned doses of DEX per cycle were 160 mg for ≤74 years and 80 mg for ≥75 years. In clinical practice, the dose of DEX is reduced unless adverse events occur. In our daily practice, the first dose of DEX was 20 mg, second dose of DEX was 8 mg, and third or later dose of DEX was 4–8 mg, independent of the incidence of adverse events. DEX was generally administered on days 1 and 15 during cycles 3–6 of DRd, and on day 1 after seven cycles of DRd. The initial dose of LEN was 25 mg/body in patients without renal insufficiency (estimated glomerular filtration rate [eGFR] ≥ 60 mL/min) independently from age and frailty and adjusted via eGFR according to the therapeutic data sheet. We did not reduce lenalidomide intentionally or in a response-adopted manner as well. When adverse events such as neutropenia and infection occurred, the LEN dose was reduced, but the DARA dose was not. Schedule adjustments for DARA and LEN were not performed even during the COVID-19 pandemic, including the COVID-19 vaccinations. Consequently, a concrete decision regarding the DRd dose depends on the physician’s choice.

### 2.4. Prophylaxis for Infection

Sulfamethoxazole-trimethoprim for *Pneumocystis jirovecii* and acyclovir for herpes zoster were administered as prophylactic agents, which is uncommon, as antifungal agents are not generally used for prophylaxis. Granular cell-stimulating factor was not administered as primary prophylaxis for neutropenia, and immunoglobulin (Ig) replacement is infrequently performed for hypoimmunoglobulinemia. However, the actual prophylaxis for infection was decided by the physician.

### 2.5. Statistical Analysis

The primary endpoint of this study was the association between the RDIs of DRd and OS. OS was calculated from the first day of DRd to the date of all-cause mortality. The secondary endpoints were TTNT, treatment response, including MRD, and safety. TTNT was calculated from the first day of DRd to the first day of the next treatment or all-cause mortality. Actuarial survival analysis was performed using the Kaplan–Meier method, and the resulting curves were compared using the log-rank test. All prognostic variables were analyzed for survival and predictive factors for OS and TTNT using Cox regression and multiple logistic regression analyses, respectively. The cumulative incidence of infection was analyzed using the Gray’s test. Multiple imputations were employed for variables with missing values, and a multivariate imputation algorithm using chained equations was used. The results of the 20 imputed datasets were combined for further analysis. All reported *p*-values were two-sided, and *p* < 0.05 was considered statistically significant. Statistical analyses were performed using EZR (Saitama Medical Center, Jichi Medical University), a graphical user interface for R (R Foundation for Statistical Computing) [17]. Specifically, it is a modified version of R Commander, which incorporates frequently used biostatistical functions.

## 3. Results

### 3.1. Patient Characteristics

In total, 111 patients with a median age of 74 years (range, 48–96 years) participated in this study. The number of patients treated with DRd as first- and second-line or later was 40 and 71, respectively. The percentages of HRCA, International Staging System (ISS) Stage 3, and performance status (PS) ≥ 2 were 23.4%, 20.7%, and 20.7%, respectively. In patients treated with DRd as second-line or later treatment, the median number of prior treatment lines was two (1–7). The percentages of exposure to PIs, refractory to PIs, exposure to LEN, refractory to LEN, and history of ASCT were 93.0%, 83.1%, 56.3%, 43.7%, and 21.1%, respectively, in patients treated with DRd as second-line or later treatment.

### 3.2. Exposure and RDI of DRd

The median number of DRd cycles was 18 (range, 1–74). The lenalidomide dose was not reduced in 47 patients. The median initial dose of lenalidomide was 10 mg; the initial dose of lenalidomide was 25 mg in five patients, 15 mg in six, 10 mg in twenty-two, 5 mg in thirteen, and 5 mg in one, with dosing every alternate day. The median number of cycles when lenalidomide was reduced for the first time was 2 (range, 1–31). Among patients in whom the initial dose of lenalidomide was reduced, 26 did not reduce additionally. The median second dose of lenalidomide was 10 mg, and the second doses of lenalidomide were 15 mg in seven patients, 10 mg in eight, 5 mg in eight, and 5 mg in three, again with dosing every alternate day. The median number of cycles when lenalidomide was reduced for the second time was 6 (range, 2–32). Finally, lenalidomide was discontinued lenalidomide because of adverse events.

Regarding dose of DEX, the median cumulative dose and number of DEX sessions during the overall DRd interval were 236 mg (range, 40–1228 mg) and 22 mg (range, 1–74 mg), respectively. Dexamethasone was not reduced for 19 patients. The median initial dose of dexamethasone was 20 mg/day; the initial doses of dexamethasone were 40 mg in two patients, 20 mg in thirteen, and 8 mg in four. The median number of cycles when dexamethasone was reduced for the first time was 1 (range: 1–14). In patients who received a reduced initial dose of dexamethasone, 52 did not receive a reduced dose of dexamethasone. The median second dose of dexamethasone was 10 mg/day; the second doses were 25, 1, and 12 mg and 18, 8, and 4 mg, respectively. The median number of cycles when dexamethasone was reduced for the second time was 4 (range, 1–13). Finally, one patient intentionally discontinued dexamethasone because of worsening diabetes.

The median RDIs of DARA, LEN, and DEX were 84.0% (range, 24.4–100), 39.4% (range, <1.0–100), and 14.6% (range, 1.7–100), respectively. Therefore, the cutoff values for the RDIs of DARA, LEN, and DEX were 90%, 40%, and 15%, respectively. The percentage of patients with a high RDI of LEN in the high RDI of DARA group was significantly higher than that in the low RDI of DARA group (*p* = 0.008). The percentage of NDMM was significantly higher in the high-RDI group than that in the low-RDI group (*p* = 0.046). Table 1 summarizes the patient characteristics between the high and low RDIs of DARA.

The percentages of patients over 75 years of age and non-IgG type in the high RDI of LEN group were higher than those in the low RDI of LEN group (*p* < 0.001 and *p* = 0.023, respectively). The percentages of age under 75 years, free light chain-λ type, and prehistory of ASCT in the low RDI of DEX group were significantly higher than in the high RDI of DEX group (*p* < 0.001, *p* = 0.050, and *p* = 0.040, respectively). Patient characteristics between the high and low RDIs of LEN and DEX are listed in Appendix A, respectively.

### 3.3. Survival Time

The median follow-up time was 26.8 months (0.7–72.6 months), and the 2-year OS and TTNT rates in the cohort were 77.7% and 63.2%, respectively. The 2-year OS and TTNT rates in the high-RDI group were significantly higher than those in the low-RDI group (91.4% vs. 66.8%, hazard ratio [HR]: 0.335, 95% confidence interval [CI]: 0.155–0.727, *p* = 0.004, Figure 1a; 77.3% vs. 51.6%, HR: 0.368, 95% CI: 0.199–0.680, *p* < 0.001, Figure 1b, respectively). The 2-year OS in the DEX group with a low RDI was higher than that in the DEX group (87.7% vs. 61.0%, HR: 0.467, 95% CI: 0.234–0.930, *p* = 0.027. Figure 1c), whereas no significant difference was observed in the 2-year TTNT between the low and high RDI of DEX groups (68.0% vs. 54.3%, HR: 0.735, 95% CI: 0.412–1.310, *p* = 0.294, Figure 1d). No significant differences were found in the 2-year OS and TTNT between the high and low RDI of LEN groups (82.2% vs. 73.7%, HR: 0.693, 95% CI: 0.347–1.387, *p* = 0.298, Figure 1e; 67.2% vs. 59.7%, HR: 0.885, 95% CI: 0.504–1.552, *p* = 0.668, Figure 1f, respectively).

In univariate analysis, a high RDI of DARA, low RDI of DEX, first-line DRd, and achievement of complete response (CR) predicted a long OS. Multivariate analysis revealed that a low RDI for DEX (HR: 0.464, 95% CI: 0.221–0.972, *p* = 0.042), first-line DRd (HR: 0.308, 95% CI: 0.098–0.968, *p* = 0.043), and CR achievement (HR: 0.259, 95% CI: 0.085–0.794; *p* = 0.021) were significant prognostic factors for OS. Concerning TTNT, a high RDI for DARA, IgG type, lactate dehydrogenase < 230 U/L, first-line DRd, and CR predicted a long TTNT. Multivariate analysis showed that a high RDI of DARA predicted long TTNT (HR: 0.503, 95% CI: 0.257–0.771, *p* = 0.044). Table 2 summarizes the OS and TTNT.

When the analyses between NDMM and RRMM were divided, the high RDI of DARA and low RDI of DEX were not associated with long OS and TTNT because the number of events for OS and TTNT was quite low (4 and 5, respectively). In the NDMM group, the RDIs of DARA, DEX, and LEN were not associated with the OS (*p* = 0.173, 0.327, and 0.211, respectively). The 3-year TTNT in the high RDI of LEN group was significantly higher than that in the low RDI of LEN group (94.4% vs. 75.7%, *p* = 0.035), whereas the RDIs of DARA and DEX were not related to TTNT (*p* = 0.190 and 0.945, respectively). However, using the multivariate analysis for prognostic factors, HRCA and age ≥75 years, in the univariate analysis, the low RDI of LEN was not a significant prognostic factor for TTNT (HR 0.394; 95%CI, 0.044–3.550; *p* = 0.406).

Among the RRMM patients, the 3-year OS in the high RDI of DARA and low RDI of DEX groups was significantly higher than that in the low RDI of DARA and high RDI of DEX groups, respectively (88.5% vs. 59.4%, *p* = 0.032, and 83.5% vs. 48.6%, *p* = 0.035). Multivariate analysis showed that a low RDI of DEX was associated with a long OS (HR, 0.276; 95%CI, 0.107–0.710, *p* = 0.008), whereas a high RDI of DARA was not related to OS (*p* = 0.881). Regarding TTNT in RRMM, the 3-year TTNT in the high RDI of DARA group was significantly longer than that in the low RDI of DARA group (68.8% vs. 41.1%, *p* = 0.014) while there was no significant correlation between the RDI of DEX and TTNT (*p* = 0.240). Using the multivariate analysis, the RDI of DARA tended to be associated with TTNT although there was no statistical significance (HR, 0.509; 95%CI, 0.240–1.080; *p* = 0.079). The RDI of LEN was not associated with OS or TTNT (*p* = 0.663 and 0.442, respectively).

### 3.4. Treatment Response

Overall, there was a very good partial response (VGPR) rate, and the CR rates were 86.9%, 42.1%, and 31.8%, respectively. The CR rate in the DARA group with high RDI was significantly higher than that in the DARA group (42.9% vs. 22.4%, *p* = 0.037). The CR and VGPR rates in the high RDI of LEN group were significantly higher than those in the low RDI of LEN group (41.5% vs. 22.2%, *p* = 0.039 and 52.8% vs. 31.5%, *p* = 0.032, respectively). The VGPR tended to be higher in the DEX group than in the high RDI of DEX group (53.8% vs. 23.8%, *p* = 0.003). The MRD assessment was performed in 29 patients who achieved CR. The numbers of MRD negativity and positivity at the first MRD assessment were 20 and 9, respectively. The MRD negativity rate of MRD-negativity was 17.9%. The MRD negativity rate in the high RDI of DARA group was significantly higher than that in the low RDI of DARA group (32.7% vs. 11.3%, *p* = 0.009). No significant association was found between the RDI of LEN and RDI of DEX or MRD-negativity (*p* = 0.243 and *p* = 0.230, respectively). Table 3 lists the treatment responses, including MRD status.

### 3.5. Safety

The safety profile was investigated among the RDIs of the DARA, LEN, and DEX groups, focusing on severe infections and neutropenia. The 30-d, 180-d, 1-year, and 2-year cumulative incidences of grade 3–5 infections were 8.1%, 19.6%, 27.2%, and 27.2%, respectively. The specific infections were as follows: pneumonia, 21; urinary tract infection, 3; febrile neutropenia, 2; cholangitis, 1; colitis, 1; postoperative osteomyelitis, 1; and septic shock, 1. The pathogens for infection were 1 alpha-streptococcus for sepsis, 1 *E. coli*, *Klebsiella oxytoca*, and *Bacteroides fragilis* for cholangitis, 1 *Klebsiella oxytoca* for pneumonia, 1 COVID-19 virus, 1 aspergillus for pneumonia, and 1 *P. jirovecii*. Pathogens were not identified in the other patients. When severe infections occurred, the number of patients with neutropenia grades 4 and 3 were four and two, respectively. The median IgG level immediately before severe infection was 545 (range, 255–5856) mg/dL, and nine patients had IgG levels < 400 mg/dL. The 2-year cumulative incidence of grade 3–5 infections in the DEX group with a high RDI was significantly higher than that in the DEX group (37.8% vs. 20.6%, *p* = 0.040, Figure 2a). The 2-year cumulative incidence of infection grades 3–5 in the high RDI of DARA group was also significantly lower than that in the low RDI of DARA group (34.8% vs. 18.8%, *p* = 0.049, Figure 2b). Poor PS and failure to achieve PR were associated with a high cumulative incidence of grade 3–5 infections (*p* = 0.001 and *p* = 0.043, respectively). In multivariate analysis, a low RDI of DEX reduced the cumulative incidence of infection (HR: 0.407, 95% CI: 0.191–0.868, *p* = 0.020), and poor PS was associated with a high cumulative infection incidence (HR: 2.852, 95% CI: 1.293–6.291, *p* = 0.009).

Four patients died from severe infections; two from bacterial pneumonia, one pneumonia caused by *P. jirovecii*, and one from postoperative osteomyelitis. The PS of the four patients who died from infection was 4. A patient who died from *P. jiroveci* pneumonia discontinued sulfamethoxazole-trimethoprim because of renal insufficiency and skin rash. A patient who died from postoperative osteomyelitis received cefazolin immediately after surgery. Two other patients who died of bacterial pneumonia did not receive antibiotics as prophylaxis. IgG levels were below 400 mg/dL in one of these four patients. Grade 3–4 neutropenia was not observed in any of these four patients.

## 4. Discussion

The RDIs of DARA in the present study were similar to those in the MAIA trial, whereas those of DEX and LEN were lower than those in the pivotal study. A high RDI for DARA and low RDI for DEX were associated with improved survival time. In addition, a low RDI for DEX was associated with a low incidence of severe infections. Therefore, a high RDI for DARA and low RDI for DEX could improve clinical outcomes in patients with MM treated with DRd.

In the MAIA trial, the RDIs for DARA, LEN, and DEX were 98.4%, 76.2%, and 84.2%, respectively [4]. In our study, the RDI for DARA was similar; however, the RDIs for LEN and DEX were lower than those reported in the MAIA trial. Owing to the management of adverse events such as neutropenia, infection, and skin rash, the RDI for LEN was low. In our daily practice, if Grade 4 neutropenia occurred or the next DRd cycle was delayed due to continuous grades 3–4 neutropenia, the LEN dose was reduced to a minimum of 5 mg every alternate day. If severe infection occurred, the doses of LEN and DEX were reduced. Meanwhile, the dose of DEX was reduced to decrease the possibility of immunosuppression by corticosteroids and incidence of several adverse events, such as infection, cataracts, diabetes, and osteoporosis, independent of the physician’s choice. However, the DARA dose was not reduced even when adverse events occurred, including infusion reactions, leading to the continuation of DARA administration without dose reduction until progressive disease (PD). Nevertheless, owing to patient choice and national holidays in daily clinical practice, the RDI in the DARA trial was lower than that of the MAIA trial because of the lengthened DARA interval.

The high DARA RDI contributed to the long TTNT and MRD negativity observed in the present study. DRd was continued until PD or unacceptable adverse events occurred. Dose adjustment of DARA was not permitted in pivotal studies. Excluding the clinical trial for post-transplantation settings, no clinical evidence was found that DARA discontinuation affected survival time, even when MRD negativity was achieved up to the present. Two phase 3 trials comparing the discontinuation of DARA-containing treatment if the MRD was negative and continuation of DARA-containing treatment if the MRD was positive are ongoing [18,19]. Therefore, continuous DARA administration until PD remains the standard of care for DARA-containing treatments, and it is key to improving clinical outcomes, even in daily practice, considering that the high RDI of DARA contributes to the long OS and sustained MRD negativity in pivotal trials [4,6,20,21].

The low RDI of DEX was also related to long OS, independent of the ISS stage and treatment response. Similar to the results of our study, in the ECOG EA03 trial, OS in the low-dose DEX arm was longer than that in the high-dose DEX arm [22]. In the TED10893 trial, a comparison between isatuximab (ISA), an anti-CD38 monoclonal antibody, plus DEX, and ISA monotherapy showed that the CD3^+^CD8^+^ T-cell counts in the ISA plus DEX group were reduced, whereas those in the ISA monotherapy group did not change [23], suggesting that DEX reduced T cells in patients treated with anti-CD38 monoclonal antibody. However, when DEX is combined with an ISA, it has no impact on T-cell receptor clonality [23], which is a predictor of therapeutic response in patients treated with an anti-CD38 monoclonal antibody [3]. Thus, a low DEX may improve clinical outcomes by preventing reduced immunological activity.

Herein, a high RDI for DEX was associated with a high cumulative incidence of infection, independent of PS and neutropenia. The association between the RDI of corticosteroids and clinical outcomes was investigated in a clinical trial of patients with MM [20,22]. In the IFM95-01 trial, a DEX-containing regimen, mainly melphalan plus dexamethasone compared to melphalan plus prednisone, had a high response rate that was associated with a higher infection incidence [24]. Likewise, the incidence of infection in the high-dose dexamethasone plus melphalan arm was significantly higher than that in the prednisone plus melphalan arm for NDMM [25]. In the ECOG EA03 trial, which compared LEN plus high- and low-dose DEX for NDMM treatment, the cumulative incidences of infection, pneumonia, and deep vein thrombosis in the high-dose DEX arm were significantly higher than those in the low-DEX arm [22]. In the IFM2017-04 trial, DARA and LEN (DR) were compared to Rd in frail patients with NDMM [26]. The incidence of grade ≥ 3 infections was identical between the DR and Rd groups, although the incidence of grade ≥ 3 neutropenia in the DR group was significantly higher than that in the Rd group, suggesting that omitting DEX contributed to a reduced incidence of more severe infections, even in patients with frailty or severe neutropenia. Thus, reducing the DEX dose may decrease the incidence of severe infections, even in patients treated with DRd.

The association between the RDI and clinical outcomes is limited to the real-world data of patients with MM. According to a Japanese, prospective, observational study of RRMM treated with elotuzumab and LEN plus DEX, a low RDI of DEX was associated with high PFS in univariate analysis. Meanwhile, no significant correlation was observed between PFS and the RDI of LEN [13], suggesting that the RDI of DEX might affect clinical efficacy in patients with MM treated with a monoclonal antibody, LEN plus DEX regimen. Dose modification of monoclonal antibodies is not recommended; therefore, the RDI for monoclonal antibodies may depend on the incidence of adverse events and modified administration schedules, such as skipping. In our study, the low RDI of DARA tended to be associated with the incidence of severe infection because DARA was skipped or discontinued if infection occurred, which could be a reasonable management strategy in daily practice.

Therefore, reducing the RDI of DEX might improve clinical outcomes and decrease the incidence of severe infections. However currently, no evidence describes how to intentionally reduce DEX during treatment, although a direct comparison of the original dose of DEX during NDMM has been presented [22,26]. Generally, the DEX dose is reduced according to protocol if adverse events occur in the clinical trials. Thus, the intentional reduction in DEX remains controversial. In this study, the low RDI of DEX was the result of its intentional reduction during DRd in daily practice. Therefore, intentional reduction in DEX might be a possible strategy, as it contributed to the clinical outcome from the efficacy and safety perspectives.

The present study has several limitations. First, some clinical data were insufficient for analysis, because the dataset was organized according to daily clinical practice. Second, the RDI was affected not only by adverse events or PD but also by the presence of national holidays and patient requests to skip treatment. Finally, this was a small-scale retrospective study; therefore, our results should be verified using another cohort or a large-scale multicenter prospective study on patients with MM treated with a reduced dose of DRd.

## 5. Conclusions

A high RDI for DARA and low RDI for DEX might be associated with a significantly improved survival time. Additionally, the high RDI of DARA contributed to the MRD negativity. In addition, a low DEX may reduce the incidence of severe infections; however, the mechanism of intentional reduction remains unclear. Thus, maintaining a high RDI for DARA and intentionally reducing the DEX dose may improve clinical outcomes during DRd treatment.

## Figures and Tables

**Figure 1 cancers-17-00470-f001:**
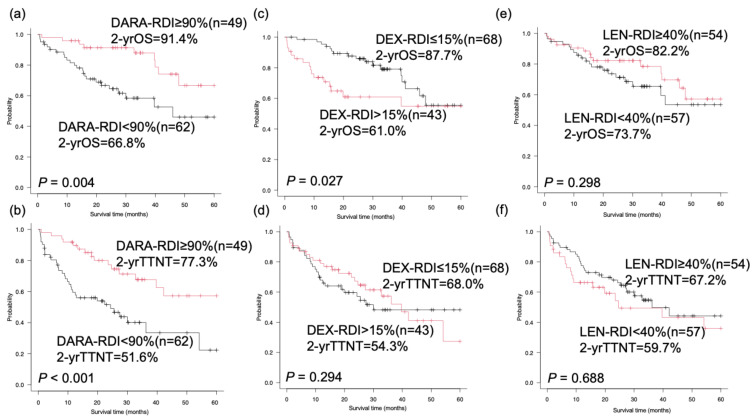
Overall survival and time to next treatment between the RDI of DRd. The TTNT was significantly longer in the high-RDI group than in the low-RDI group. The OS of the DEX group with a low RDI was significantly longer than that of the DEX group with a high RDI. (**a**) OS in the high and low RDI of DARA groups, (**b**) OS in the high and low RDI of LEN groups, (**c**) OS in the high and low RDI of DEX groups, (**d**) TTNT in the high and low RDI of DARA groups, (**e**) TTNT in the high and low RDI of LEN groups, and (**f**) TTNT in the high and low RDI of DEX groups. OS, overall survival; TTNT, time to next treatment; RDI, relative dose intensity; DARA, daratumumab; LEN, lenalidomide; DEX, dexamethasone.

**Figure 2 cancers-17-00470-f002:**
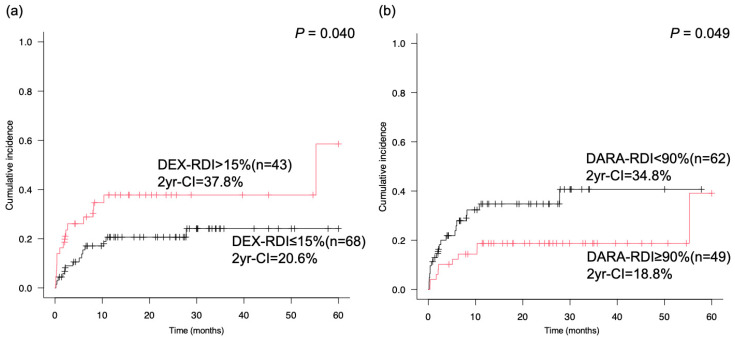
Cumulative incidence of infection grades 3–5 between the high and low RDI of DEX and low and high RDI of DARA. The CI of severe infection in the high RDI of DEX was significantly higher than that in the low RDI of DEX. (**a**) The 2-year CI of infection grades 3–5 in the high and low RDI of DEX and (**b**) low RDI of DARA group. CI, cumulative incidence; RDI, relative dose intensity; DARA, daratumumab; DEX, dexamethasone.

**Table 1 cancers-17-00470-t001:** Patient characteristics.

	High RDI of DARA (*n* = 49)	Low RDI of DARA (*n* = 62)	*p*-Value
Median age	median 74 years (range, 48–96)		
Mean age (years)	75 ± 7	73 ± 8	0.241
≥75 years	26	28	0.448
<75 years	23	34	
Sex			
male	27	36	0.848
female	22	26	
ECOG performance status			
0, 1	43	45	0.061
2, 3, 4	6	17	
Type of monoclonal protein			
IgG	23	31	0.849
non-IgG	26	31	
Type of free light chain			
kappa	29	35	0.848
lambda	20	27	
ISS			
stage 1, 2	35	41	0.472
stage 3	8	15	
unknown	6	6	
eGFR			
≥60 mL/min	22	37	0.433
<60 mL/min	24	24	
unknown	3	1	
Serum LDH level			
Mean serum LDH	227.1 ± 164.6	230.9 ± 91.7	0.879
≥230 U/L	15	24	0.427
<230 U/L	34	38	
High-risk cytogenetic abnormality			
yes	9	17	0.147
no	28	23	
unknown	12	22	
Treatment line of DRd			
first-line	23	17	0.046
second-line or later	26	45	
High RDI of lenalidomide			
yes	31	23	0.008
no	18	39	
Low RDI of dexamethasone			
yes	28	40	0.441
no	21	22	
Proteasome inhibitor exposure			
yes	23	43	0.348
no	3	2	
Proteasome inhibitor refractory			
yes	20	39	0.335
no	6	6	
Lenalidomide exposure			
yes	15	25	0.999
no	11	20	
Lenalidomide refractory			
yes	10	21	0.621
no	16	24	
ASCT prehistory			
yes	5	10	0.999
no	21	35	

RDI, relative dose intensity; DARA, daratumumab; eGFR, estimated glomerular filtration rate; DRd, daratumumab, lenalidomide plus dexamethasone; ASCT, autologous stem cell transplantation.

**Table 2 cancers-17-00470-t002:** Univariate and multivariate analysis of overall survival and time to next treatment.

	Univariate		Multivariate Model			Univariate		Multivariate Model		
	2-Year OS	*p*-Value	Hazard Ratio	95% CI	*p*-Value	2-Year TTNT	*p*-Value	Hazard Ratio	95% CI	*p*-Value
High RDI of daratumumab										
yes	91.4%	0.004	0.520	0.218–1.238	0.133	77.3%	<0.001	0.503	0.257–0.771	0.044
no	66.8%		reference			51.6%		reference		
High RDI of lenalidomide										
yes	82.2%	0.298				67.2%	0.668			
no	73.7%					59.7%				
Low RDI of dexamethasone										
yes	87.7%	0.027	0.464	0.221–0.972	0.042	68.0%	0.294			
no	61.0%		reference			54.3%				
Age										
≥75 years	70.7%	0.195				56.6%	0.316			
<75 years	84.6%					69.7%				
Sex										
male	74.6%	0.756				63.8%	0.946			
female	82.1%					62.8%				
ECOG performance status										
0, 1	81.3%	0.085				65.6%	0.410			
2, 3, 4	64.1%					53.5%				
Type of monoclonal protein										
IgG	82.7%	0.185				74.9%	0.005	0.395	0.202–0.771	0.007
non-IgG	72.6%					51.3%		reference		
Type of free light chain										
kappa	86.4%	0.060				72.6%	0.158			
lambda	66.8%					51.5%				
ISS										
stage 1, 2	80.8%	0.030	reference			63.7%	0.052			
stage 3	60.9%		1.891	0.786–4.551	0.146	46.4%				
eGFR										
≥60 mL/min	78.8%	0.430				63.8%	0.757			
<60 min/mL	76.1%					60.8%				
LDH										
≥230 U/L	68.7%	0.069				51.1%	0.005	2.101	1.139–3.876	0.017
<230 U/L	82.3%					69.3%		reference		
High-risk cytogenetic abnormality										
yes	79.4%	0.679				67.4%	0.493			
no	81.0%					71.1%				
Treatment line of DRd										
First-line	91.6%	0.018	0.308	0.098–0.961	0.043	86.1%	<0.001	0.289	0.119 –0.704	0.006
Second-line or later	70.6%		reference			51.5%		reference		
Proteasome inhibitor exposure										
yes	71.5%	0.614				51.0%	0.696			
no	60.0%					60.0%				
Proteasome inhibitor refractory										
yes	72.0%	0.733				50.9%	0.683			
no	62.3%					55.6%				
Lenalidomide exposure										
yes	68.8%	0.404				45.1%	0.347			
no	73.0%					59.3%				
Lenalidomide refractory										
yes	62.5%	0.116				34.8%	0.120			
no	76.8%					63.7%				
ASCT prehistory										
yes	78.3%	0.567				58.7%	0.998			
no	68.5%					49.5%				
Treatment response										
CR or better	100.0%	<0.001	0.259	0.085–0.794	0.021	96.8%	<0.001	0.172	0.763–0.409	<0.001
VGPR or worse	67.7%		reference			48.9%		reference		

OS, overall survival; TTNT, time to next treatment; RDI, relative dose intensity; ISS, International Staging System; eGFR, estimated glomerular filtration rate; LDH, lactate dehydrogenase; DRd, daratumumab, lenalidomide plus dexamethasone; ASCT, autologous stem cell transplantation; CR, complete response; VGPR, very good partial response.

**Table 3 cancers-17-00470-t003:** Treatment response.

	High RDI of DARA (*n* = 49)	Low RDI of DARA (*n* = 62)	*p*-Value	High RDI of LEN (*n* = 54)	Low RDI of LEN (*n* = 57)	*p*-Value	Low RDI of DEX (*n* = 68)	High RDI of DEX (*n* = 43)	*p*-Value
Overall response rate	93.9%	81.0%	0.082	87.2%	87.0%	0.999	89.2%	83.3%	0.394
Very good partial response rate	51.0%	34.5%	0.116	52.8%	31.5%	0.032	53.8%	23.8%	0.003
Complete response rate	42.9%	22.4%	0.037	41.5%	22.2%	0.039	38.5%	21.4%	0.089
MRD negative rate	32.7%	11.3%	0.009	25.9%	15.8%	0.243	29.3%	14.0%	0.230

RDI, relative dose intensity; DARA, daratumumab; LEN, lenalidomide; DEX, dexamethasone; MRD, minimal residual disease.

## Data Availability

Data are contained within the article.

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
