# Peer review of "Relative Dose Intensity of Daratumumab, Lenalidomide, and Dexamethasone in Multiple Myeloma"

_cancers, 2025, doi:10.3390/cancers17030470_

Round 1
Reviewer 1 Report
Comments and Suggestions for Authors
This is an important paper that aims to study the relative dose intensity effect of the DRd triple combination used to treat multiple myeloma. Though the data are retrospective in nature, they nevertheless call attention to the importance of high Dara versus low Dexa intensities throughout myeloma therapy. Time to next treatment, infection rates as well as OS were both similarly affected.
In Simple Summary, line 12 and Abstract line 21: it is an exaggeration to call DRd as the standard of care, it is certainly one of the standards of care in multiple clinical scenarios in myeloma. The opposite effects of Dara and Dex dose intensities on infection rates should be also mentioned in the abstract, as the Dara positive effect is a novel observation.
Dose adjustment, line 78: what was the protocol used for Dara dose omission during COVID-19 infection and COVID vaccination? It is customary to omit anti-CD38 during COVID-19 infection. What was the protocol followed by the authors? Dara dosing (omission) is mentioned in the Discussion section but should rather be detailed here in the Methods sections. DEX dose reductions are detailed, but Dara dose modifications are not. How was the Len dose modification? If the renal function of the patient improved, did the authors adjust (increased) the lenalidomide dose?
It is not unusual to see progression events soon after planned – according to label - Dara dose reductions at week 9 and month 5. Did the authors also observe this phenomenon?
Figure 1 would be better observable if presented not in grayscale but with the progression curves drawn in different colors.
Author Response
This is an important paper that aims to study the relative dose intensity effect of the DRd triple combination used to treat multiple myeloma. Though the data are retrospective in nature, they nevertheless call attention to the importance of high Dara versus low Dexa intensities throughout myeloma therapy. Time to next treatment, infection rates as well as OS were both similarly affected.
->Thank you very much for taking the time to review this manuscript. Please find the detailed responses below and corresponding revisions/corrections highlighted/in track changes in the re-submitted files.
In Simple Summary, line 12, and Abstract line 21: it is an exaggeration to call DRd as the standard of care, it is certainly one of the standards of care in multiple clinical scenarios in myeloma. The opposite effects of Dara and Dex dose intensities on infection rates should be also mentioned in the abstract, as the Dara positive effect is a novel observation.
Thank you for the suggestion. We revised the expression for DRd from “the standard of care” to “one of the standards of care” in simple summary (line 12) and abstract (line 20-21). And “The high RDI of DARA and low RDI of DEX reduced the incidence of severe infections.” was added in simple summary, line 16-17, and abstract, line 32-33.
Dose adjustment, line 78: what was the protocol used for Dara dose omission during COVID-19 infection and COVID vaccination? It is customary to omit anti-CD38 during COVID-19 infection. What was the protocol followed by the authors? Dara dosing (omission) is mentioned in the Discussion section but should rather be detailed here in the Methods sections. DEX dose reductions are detailed, but Dara dose modifications are not. How was the Len dose modification? If the renal function of the patient improved, did the authors adjust (increased) the lenalidomide dose?
Thank you for the suggestion. We did not adjust (omit) the dose of DARA, even during the COVID-19 pandemic; however, we adjusted the DARA administration schedule according to the patient’s will, national holiday, and incidence of adverse events such as infection and cytopenia. We did not adjust the dose of LEN even during the COVID-19 pandemic, and the initial dose of LEN was 25 mg/body in the patients without renal insufficiency (eGFR60mL/min) independently from age and frailty, and adjusted via eGFR according to the therapeutic data sheet. We also adjusted the schedule of LEN because of the national holiday and incidence of adverse events. For instance, when the interval of DRd was 5 weeks due to a national holiday, LEN was administered for 21 d and off for 14 d. The LEN dose was increased if adverse events did not occur in the patients who received 10 mg/day of the initial LEN dose due to an eGFR rate of 30–60 or renal insufficiency improved during DRd, but the number of these cases was quite low. We have added an explanation of the schedule adjustment for DARA and LEN as follows:
Line 87-90.
The initial dose of LEN was 25 mg/body in patients without renal insufficiency (estimated glomerular filtration rate [eGFR]60mL/min) independently from age and frailty, and adjusted via eGFR according to the therapeutic data sheet.
Line 91-93.
Schedule adjustments for DARA and LEN were not performed even during the COVID-19 pandemic, including the COVID-19 vaccinations.
It is not unusual to see progression events soon after planned – according to label - Dara dose reductions at week 9 and month 5. Did the authors also observe this phenomenon?
Thank you for the suggestion. Adjusting the schedule for DARA within the initial six cycles was uncommon in our practice, even during the COVID-19 pandemic. Adjustment of the DARA schedule from weekly to biweekly in the second cycle was performed for a few patients with severe renal insufficiency, depending on hemodialysis and difficulty visiting the hospital frequently due to social issues.
Figure 1 would be better observable if presented not in grayscale but with the progression curves drawn in different colors.
Thank you for your insightful suggestion. We have revised the figures to two colors (black and red).
Reviewer 2 Report
Comments and Suggestions for Authors
This paper from Suzuki et al covers the analysis of Jikei University Kashiwa Hospital on treating multiple myeloma patients using DRd and the potential impact of the relative dose intensity on clinical outcomes. Although dose intensity has been a matter of debate in several haematolymphoid neoplasms, some practical conclusions on the optimal use of several drugs have been obtained in the last decades. For instance, as the authors correctly state in the text, lenalidomide dose in older patients.
The manuscript in its present form is easy to read, and quite logical in its intentions. However, there are major issues within.
First, data collection and tabulation. Due to the retrospective nature of the present work, data should be verified for consistency. The authors indicate some data are missing and they can act as confounders. Tables and text must be edited for increased clarity.
Second, and most worrisome. The authors provide a mix of patients characteristics (first line, transplant, second & ulterior lines) and possibly different therapeutic combinations, which may preclude to obtain adequate conclusions.
Third, following the previous issue, the results in the multivariate analysis clearly reflect what is already known in MM treatment.
Fourth, although the way the authors have calculated the DRI seems clear, maybe indicating the planned dose and administrated dose would be useful. Dexamethasone dose is age-dependent, so expressing DRI as a percentage can mislead the dose in younger patients who are able to tolerate higher doses.
Fifth, have the authors explored the impact of dose reduction in early versus late treatment phases?
Sixth, DRI seems not to be a modifiable agent. How do the authors plan to translate this into practice?
Author Response
This paper from Suzuki et al covers the analysis of Jikei University Kashiwa Hospital on treating multiple myeloma patients using DRd and the potential impact of the relative dose intensity on clinical outcomes. Although dose intensity has been a matter of debate in several haematolymphoid neoplasms, some practical conclusions on the optimal use of several drugs have been obtained in the last decades. For instance, as the authors correctly state in the text, lenalidomide dose in older patients.
Thank you for reviewing the manuscript. Please find the detailed responses below and corresponding revisions/corrections highlighted/in track changes in the re-submitted files.
The manuscript in its present form is easy to read, and quite logical in its intentions. However, there are major issues within.
First, data collection and tabulation. Due to the retrospective nature of the present work, data should be verified for consistency. The authors indicate some data are missing and they can act as confounders. Tables and text must be edited for increased clarity.
Thank you for your valuable suggestion. We reanalyzed the multivariate analysis by multivariate imputation using chained equations to analyze the datasets that included missing data. We added the explanation for the multivariate analysis by the multivariate imputation using chained equations in the “Methods” part and deleted the column for “unknown” in the eGFR, ISS and high-risk cytogenetic abnormality and changed the results for the multivariate analysis for OS in Table 2 as follows.
Line 111-113.
Multiple imputations were employed for variables with missing values, and a multivariate imputation algorithm using chained equations was used. The results of the 20 imputed datasets were combined for further analysis.
Line 172-175.
Multivariate analysis revealed that a low RDI for DEX (HR: 0.464, 95% CI: 0.221–0.972, P = 0.042), first-line DRd (HR: 0.308, 95% CI: 0.098–0.968, P = 0.043), and CR achievement (HR: 0.259, 95% CI: 0.085–0.794; P = 0.021) were significant prognostic factors for OS.
Second, and most worrisome. The authors provide a mix of patients characteristics (first line, transplant, second & ulterior lines) and possibly different therapeutic combinations, which may preclude to obtain adequate conclusions.
Thank you for the suggestion. This retrospective trial included patients treated for DRd under various conditions. However, the DRd schedule was consistent for all patients; therefore, we planned to analyze the impact of RDI on DRd in patients with both NDMM and RRMM. Additionally, in the multivariate analysis, the impact of a low RDI of DEX was observed including NDMM as a prognostic factor, suggesting that the low RDI of DEX was significantly associated with a long OS independent of the therapeutic line of DRd. When the analyses between NDMM and RRMM were divided, the high RDI of DARA and low RDI of DEX were not associated with long OS or TTNT due to the low number of events for OS and TTNT (4 and 5, respectively). Meanwhile, among the patients with RRMM, a low RDI of DEX was associated with long OS using multivariate analysis, whereas a high RDI of DARA was related to long TTNT using univariate analysis. Finally, a low RDI of DEX was not associated with the incidence of severe infections significantly focusing on the patient with RRMM (P = 0.055). We do agree with the reviewer’s opinion, so we have changed the expression of the conclusion to reduce the intensity of the conclusion as follows.
Line 327-328.
A high RDI of DARA and a low RDI of DEX were associated with significantly improved survival time.
->
A high RDI for DARA and a low RDI for DEX might be associated with a significantly improved survival time.
Third, following the previous issue, the results in the multivariate analysis clearly reflect what is already known in MM treatment.
Thank you for your important opinion. We totally agreed with the reviewer’s opinion. Therefore, we discussed the benefit of reduced dose of DEX using the previous clinical trials, such as ECOG EA03 and IFM2017-04 trials in the “Discussion” part.
Fourth, although the way the authors have calculated the DRI seems clear, maybe indicating the planned dose and administrated dose would be useful. Dexamethasone dose is age-dependent, so expressing DRI as a percentage can mislead the dose in younger patients who are able to tolerate higher doses.
Thank you for the suggestion. The RDI was calculated using the planned initial doses of LEN and DEX. Therefore, the denominator of DEX differed depending on the age at which its RDI was calculated. Similarly, the denominator of LEN differed depending on its renal function when DRd was initiated.
Fifth, have the authors explored the impact of dose reduction in early versus late treatment phases?
Thank you for your valuable suggestion. When the revised manuscript was prepared, we analyzed the impact of RDI on DRd divided onto patients with NDMM and RRMM in response to the second comment.
Sixth, DRI seems not to be a modifiable agent. How do the authors plan to translate this into practice?
Thank you for your comment. We usually administer DARA and LEN followed by the pivotal trial, but reduce DEX intentionally, as well as the method used in this study. However, considering the results of this study, the dose of LEN may remain controversial, especially in elderly or frail patients. Meanwhile, we lack sufficient evidence on how to initially reduce the dose of LEN; therefore, we do not plan to change the dose/schedule of DRd for both NDMM and RRMM.
Reviewer 3 Report
Comments and Suggestions for Authors
The authors presented data of relative dose intensity of DRd regimen in a real-world, retrospective analysis. This data is potentially important in guidling clinical practice. In fact, I think it is an important reference in designing treatment regimen for multiple myeloma in the future. Despite such importance, some potential drawbacks and concerns are present.
1# The analysis between groups is based on dose intensity. However, in a retrospective analysis, there were reasons for dose intensity to be different. The ECOG performance status, lenalidomide dose density and the line of treatment appeared to be a little different. It is likely low dose intensity resulted from the poor conditions of patients, which is the main factor leading to less favorable outcome, rather than the dose intensity.
2. The author may perform a case-control analysis, choosing conditions comparable between high and low dose density (for example, the same line of treatment) to confirm the impact of dose density.
3. For numeric variable, the analysis should be t-test, not categorial. Age, LDH are numeric variables. Categorial analysis is the wrong method and the cutoff (age 75, LDH 230) appear to be arbitrary.
4. A retrospetive data should be interpreted with caution and the limit should be clearly described. The authors should re-phrase some descriptions and downplay the statement.
5. The principle of dose modification should be described. It is expected in a retrospective data, such dose modification is the discretion of treating physicians. Even so, the reason for dose reduction (including dexamethasone) should be analysed and presented.
6. The authors stated a lower steroid dose is associated with less toxicity. This statement should be supported by solid data, either from this study (better) or quotation from other published literature.
Author Response
The authors presented data of relative dose intensity of DRd regimen in a real-world, retrospective analysis. This data is potentially important in guidling clinical practice. In fact, I think it is an important reference in designing treatment regimen for multiple myeloma in the future. Despite such importance, some potential drawbacks and concerns are present.
Thank you for reviewing the manuscript. Please find the detailed responses below and corresponding revisions/corrections highlighted/in track changes in the re-submitted files.
1# The analysis between groups is based on dose intensity. However, in a retrospective analysis, there were reasons for dose intensity to be different. The ECOG performance status, lenalidomide dose density and the line of treatment appeared to be a little different. It is likely low dose intensity resulted from the poor conditions of patients, which is the main factor leading to less favorable outcome, rather than the dose intensity.
Thank you for your comment. This retrospective trial included patients treated for DRd under various conditions. However, the DRd schedule was consistent for all patients in this retrospective study. Poor PS could not affect OS, TTNT, or the incidence of severe infections in this study. The initial dose of LEN was 25 mg/body in patients without renal insufficiency (eGFR 60mL/min), independent from age and frailty, and adjusted via eGFR according to the therapeutic data sheet. The impact of a low RDI of DEX was also observed in the multivariate analysis, including NDMM as a prognostic factor, suggesting that the low RDI of DEX was statistically associated with long OS independent of the therapeutic line of DRd. Therefore, the high RDI of DARA and low RDI of DEX had a significant clinical impact, independent of PS, LEN dose, and disease status.
- The author may perform a case-control analysis, choosing conditions comparable between high and low dose density (for example, the same line of treatment) to confirm the impact of dose density.
Thank you for your valuable suggestion. The DRd schedule was consistent for all patients; therefore, we analyzed the impact of RDI on DRd in patients with both NDMM and RRMM. Additionally, in the multivariate analysis, the impact of a low RDI of DEX was observed, including NDMM as a prognostic factor, suggesting that the low RDI of DEX was statistically associated with long OS independent of the therapeutic line of DRd. When the analyses between NDMM and RRMM were divided, the high RDI of DARA and low RDI of DEX were not associated with long OS or TTNT due to the low number of events for OS and TTNT (4 and 5, respectively). Meanwhile, among the patients with RRMM, a low RDI of DEX was associated with long OS using multivariate analysis, whereas a high RDI of DARA was related to long TTNT using univariate analysis. Finally, a low RDI of DEX was not associated with the incidence of severe infections significantly focusing the patients with RRMM (P = 0.055).
- For numeric variable, the analysis should be t-test, not categorial. Age, LDH are numeric variables. Categorial analysis is the wrong method and the cutoff (age 75, LDH 230) appear to be arbitrary.
Thank you for the suggestion. We decided on the cutoff for age and LDH level based on the median age in this study and upper normal limit in our hospital. Additionally, we analyzed the mean age and LDH between the high and low RDI of DARA; the mean age between the high and low RDI of DARA were 75 and 73 years, respectively (P = 0.879), and the mean LDH between the high and low RDI of DARA were 227.1 and 230.9 U/L, respectively (P = 0.241). Thus, no significant difference were observed in age or LDH between the high and low RDI DARA groups using either Fisher’s exact test or t-test. Meanwhile, the mean age and serum LDH levels were significantly higher in the high RDI of LEN and high RDI of DEX groups than in the low RDI of LEN and low RDI of DEX groups using the t-test, similar to the results using Fisher’s exact test. We have added the mean age and serum LDH level in Table 1 and Supplementary Tables 1 and 2.
- A retrospetive data should be interpreted with caution and the limit should be clearly described. The authors should re-phrase some descriptions and downplay the statement.
Thank you for the suggestion, with which we agree. Therefore, we revised the sentence as follows:
Line 322-325.
Finally, as this was a small-scale retrospective study; therefore, our results should be verified using another cohort or a large-scale multicenter prospective study on patients with MM treated with a reduced dose of DRd.
- The principle of dose modification should be described. It is expected in a retrospective data, such dose modification is the discretion of treating physicians. Even so, the reason for dose reduction (including dexamethasone) should be analyzed and presented.
Thank you for the suggestion. We usually administer DARA and LEN, followed by a pivotal trial, but intentionally reduce DEX. We did not adjust the dose of DARA, even during the COVID-19 pandemic. Therefore, we adjusted the DARA administration schedule according to the patient’s will, national holiday, and incidence of adverse events such as infection and cytopenia. We did not adjust the dose of LEN even during the COVID-19 pandemic, including vaccination for COVID-19. The initial dose of LEN was 25 mg/body in the patients without renal insufficiency (eGFR60mL/min), independent from age and frailty, and adjusted via eGFR according to the therapeutic data sheet. We also adjusted the schedule of LEN because of the national holiday and incidence of adverse events. For instance, when the interval of DRd was 5 weeks due to a national holiday, LEN was administered for 21 d and off for 14 d. The dose of LEN was increased if adverse events did not occur in the patients who received 10 mg/day of the initial dose of LEN due to an eGFR level of 30–60 or renal insufficiency improved during DRd, but the number of these cases was quite low. Finally, the dose of DEX was reduced unless adverse events occurred as mentioned in line 82-86; The first dose of DEX was 20 mg, second dose of DEX was 8 mg, and third or later doses of DEX was 4–8 mg independently from incidence of adverse events. DEX was generally administered on days 1 and 15 during cycles 3–6 of DRd, and on day 1 after seven cycles of DRd. We have added an explanation of the schedule adjustment for DARA and LEN as follows.
Line 87-90.
The initial dose of LEN was 25 mg/body in patients without renal insufficiency (estimated glomerular filtration rate [eGFR]60mL/min) independently from age and frailty, and adjusted via eGFR according to the therapeutic data sheet.
Line 91-93.
Schedule adjustments for DARA and LEN were not performed even during the COVID-19 pandemic, including the COVID-19 vaccinations.
- The authors stated a lower steroid dose is associated with less toxicity. This statement should be supported by solid data, either from this study (better) or quotation from other published literature.
Thank you for the suggestion. We have described several lines of evidence regarding the low incidence of adverse events, such as infection, pneumonia, and DVT, using the results of the ECOG EA03 and IFM2017-04 trials in lines 274-299. Additionally, the IFM95-01 trial revealed that the DEX-containing regimen had a high response rate, but was associated with a higher incidence of infection than melphalan plus prednisone. We added the results of the IFM95-01 trial as follows:
Line 284-286.
In the IFM95-01 trial, a DEX-containing regimen, mainly melphalan plus dexamethasone compared to melphalan plus prednisone, had a high response rate that was associated with a higher infection incidence.
Lines 455-460.
- Facon, T.; Mary, JY.; Pégourie, B.; Attal, M.; Renaud, M.; Sadoun, A.; Voillat, L.; Dorvaux, V.; Hulin, C.; Lepeu, G.; Harousseau, JL.; Eschard, JP.; Ferrant, A.; Blanc, M.; Maloisel, F.; Orfeuvre, H.; Rossi, JF.; Azaïs, I.; Monconduit, M.; Collet, P.; Anglaret, B.; Yakoub-Agha, I.; Wetterwald, M.; Eghbali, H.; Vekemans, MC.; Maisonneuve, H.; Troncy, J.; Grosbois, B.; Doyen, C.; Thyss, A.; Jaubert, J.; Casassus, P.; Thielemans, B.; Bataille, R; Intergroupe Francophone du Myélome (IFM) group. Dexamethasone-Based Regimens versus melphalan-prednisone for elderly patients with multiple myeloma eligible for high-dose therapy. Blood 2006, 107 (4), 1292–1298. https://doi.org/10.1182/blood-2005-04-1588.
Round 2
Reviewer 2 Report
Comments and Suggestions for Authors
The authors have provided information lacking in the previous version. Interestingly, they have used a multiple imputation by chained equations (MICE) protocol to compensate missing results. However, this assumes the missing data have occurred at random, which may not be the case due to the retrospective nature of this work.
Also, the dose reduction referers only to the total dose, therefore the question if dose reduction has any role in early (for instance, first months of treatment) or late (for instance, after acheaving a response) has not been answered, which could be an interesting practice point.
Also, the differentiation of NDMM and RRMM patients is neccesary, as results differ. Due to the low number of patients, this is a problem with this dataset, but we encourage the authors to increase their series and provide separate reports.
Finally, although partial, these results confirm what already has been known in multiple myeloma community, which can be a valuable repetition.
Author Response
The authors have provided information lacking in the previous version. Interestingly, they have used a multiple imputation by chained equations (MICE) protocol to compensate missing results. However, this assumes the missing data have occurred at random, which may not be the case due to the retrospective nature of this work.
Also, the dose reduction referers only to the total dose, therefore the question if dose reduction has any role in early (for instance, first months of treatment) or late (for instance, after acheaving a response) has not been answered, which could be an interesting practice point.
Thank you for your important suggestion. We agree with your point, and have consequently added information regarding the timing of reducing the dose of LEN and DEX.
We reduced lenalidomide because of adverse events, mainly infection, neutropenia, and skin rash, but did not reduce lenalidomide intentionally or in a response-adopted manner. The lenalidomide dose was not reduced in 47 patients. The median initial dose of lenalidomide was 10 mg; the initial dose of lenalidomide was 25 mg in five patients, 15 mg in six, 10 mg in 22, 5 mg in 13, and 5 mg in one, with dosing every alternate day. The median number of cycles when lenalidomide was reduced for the first time was 2 (range, 1 – 31). Among patients in whom the initial dose of lenalidomide was reduced, 26 did not reduce additionally. The median second dose of lenalidomide was 10 mg, and the second doses of lenalidomide were 15 mg in seven patients, 10 mg in eight, 5 mg in eight, and 5 mg in three, again with dosing every alternate day. The median number of cycles when lenalidomide was reduced for the second time was 6 (range, 2 – 32). Finally, lenalidomide was discontinued lenalidomide because of adverse events.
We intentionally reduced dexamethasone dose, not owing to adverse events, in a response-adopted manner. After the 3rd cycle of DRd, dexamethasone was omitted on the day daratumumab was not administered. Dexamethasone was not reduced for 19 patients. The median initial dose of dexamethasone was 20 mg/day; the initial doses of dexamethasone were 40 mg in two patients, 20 mg in 13, and 8 mg in 4. The median number of cycles when dexamethasone was reduced for the first time was 1 (range: 1 – 14). In patients who received a reduced initial dose of dexamethasone, 52 did not receive a reduced dose of dexamethasone. The median second dose of dexamethasone was 10 mg/day; the second doses were 25, 1, 12 mg and 18, 8, and 4 mg, respectively. The median number of cycles when dexamethasone was reduced for the second time was 4 (range, 1 – 13). Finally, one patient intentionally discontinued dexamethasone because of worsening diabetes.
We added the concrete the timing for reduction of Len and Dex in the results as below.
Line 90 – 91.
We did not reduce lenalidomide intentionally or in a response-adopted manner as well.
Line 131 – 152.
Forty-seven patients did not reduce LEN. Median initial dose of LEN was 10mg for them; the initial dose of LEN was 5 of 25mg, 6 of 15mg, 22 of 10mg, 13 of 5mg, and 1 of 5mg every other day. Median number of cycles when LEN was reduced at the first time was 2 (range 1 – 31). In the patients reducing initial dose of LEN, 26 patients did not reduce LEN additionally. Median second dose of LEN was 10mg for them; the second dose of LEN was 7 of 15mg, 8 of 10mg, 8 of 5mg, 3 of 5mg every other day. Median number of cycles when LEN was reduced at the second time was 6 (range 2 – 32). Finally, fifteen patients discontinued LEN due to adverse events.
Regarding dose of DEX, the median cumulative dose and number of DEX sessions during the overall DRd interval were 236 mg (range, 40 – 1228 mg) and 22 mg (range, 1 – 74 mg), respectively. Dexamethasone was not reduced for 19 patients. The median initial dose of dexamethasone was 20 mg/day; the initial doses of dexamethasone were 40 mg in two patients, 20 mg in 13, and 8 mg in 4. The median number of cycles when dexamethasone was reduced for the first time was 1 (range: 1 – 14). In patients who received a reduced initial dose of dexamethasone, 52 did not receive a reduced dose of dexamethasone. The median second dose of dexamethasone was 10 mg/day; the second doses were 25, 1, 12 mg and 18, 8, and 4 mg, respectively. The median number of cycles when dexamethasone was reduced for the second time was 4 (range, 1 – 13). Finally, one patient intentionally discontinued dexamethasone because of worsening diabetes.
Also, the differentiation of NDMM and RRMM patients is neccesary, as results differ. Due to the low number of patients, this is a problem with this dataset, but we encourage the authors to increase their series and provide separate reports.
Thank you for your important suggestion. We agree with your point suggesting analyzing NDMM and RRMM separately.
When the analyses between NDMM and RRMM were divided, the high RDI of DARA and low RDI of DEX were not associated with long OS and TTNT because the number of events for OS and TTNT was quite low (4 and 5, respectively). In the NDMM group, the RDI of DARA, DEX, and LEN were not associated with the OS (P = 0.173, 0.327, and 0.211, respectively). The 3-year TTNT in the high RDI of LEN group was significantly higher than that in the low RDI of LEN group (94.4% vs. 75.7%, P = 0.035), whereas the RDI of DARA and DEX were not related to TTNT (P = 0.190 and 0.945, respectively). However, using the multivariate analysis for prognostic factors, HRCA and age75 years, in the univariate analysis, the low RDI of LEN was not a significant prognostic factor for TTNT (HR 0.394; 95%CI, 0.044 – 3.550; P =0.406).
Among the RRMM patients, the 3-year OS in the high RDI of DARA and low RDI of DEX groups was significantly higher than that in the low RDI of DARA and high RDI of DEX groups, respectively (88.5% vs 59.4%, P = 0.032, and 83.5% vs 48.6%, P = 0.035). Multivariate analysis showed that a low RDI of DEX was associated with a long OS (HR, 0.276; 95%CI, 0.107 – 0.710, P = 0.008), whereas a high RDI of DARA was not related to OS (P = 0.881). Regarding TTNT in RRMM, the 3-year TTNT in the high RDI of DARA group was significantly longer than that in the low RDI of DARA group (68.8% vs 41.1%, P=0.014) while there was no significant correlation between the RDI of DEX and TTNT (P= 0.240). Using the multivariate analysis, the RDI of DARA tended to be associated with TTNT although there was no statistical significance (HR, 0.509; 95%CI, 0.240 – 1.080; P = 0.079). The RDI of LEN was not associated with OS or TTNT (P = 0.663 and 0.442, respectively).
We added the results in the NDMM and RRMM groups in the results as below.
Line 205 – 225.
When the analyses between NDMM and RRMM were divided, the high RDI of DARA and low RDI of DEX were not associated with long OS and TTNT because the number of events for OS and TTNT was quite low (4 and 5, respectively). In the NDMM group, the RDI of DARA, DEX, and LEN were not associated with the OS (P = 0.173, 0.327, and 0.211, respectively). The 3-year TTNT in the high RDI of LEN group was significantly higher than that in the low RDI of LEN group (94.4% vs. 75.7%, P = 0.035), whereas the RDI of DARA and DEX were not related to TTNT (P = 0.190 and 0.945, respectively). However, using the multivariate analysis for prognostic factors, HRCA and age75 years, in the univariate analysis, the low RDI of LEN was not a significant prognostic factor for TTNT (HR 0.394; 95%CI, 0.044 – 3.550; P =0.406).
Among the RRMM patients, the 3-year OS in the high RDI of DARA and low RDI of DEX groups was significantly higher than that in the low RDI of DARA and high RDI of DEX groups, respectively (88.5% vs 59.4%, P = 0.032, and 83.5% vs 48.6%, P = 0.035). Multivariate analysis showed that a low RDI of DEX was associated with a long OS (HR, 0.276; 95%CI, 0.107 – 0.710, P = 0.008), whereas a high RDI of DARA was not related to OS (P = 0.881). Regarding TTNT in RRMM, the 3-year TTNT in the high RDI of DARA group was significantly longer than that in the low RDI of DARA group (68.8% vs 41.1%, P=0.014) while there was no significant correlation between the RDI of DEX and TTNT (P= 0.240). Using the multivariate analysis, the RDI of DARA tended to be associated with TTNT although there was no statistical significance (HR, 0.509; 95%CI, 0.240 – 1.080; P = 0.079). The RDI of LEN was not associated with OS or TTNT (P = 0.663 and 0.442, respectively).
Finally, although partial, these results confirm what already has been known in multiple myeloma community, which can be a valuable repetition.
Thank you for your warm comments. We also agree with you completely.
Reviewer 3 Report
Comments and Suggestions for Authors
Three issues remain.
1# Regardless of analysis method, NDMM and RRMM are different polulations and should not be pooled together for such a study.
2# The authors stated the DRd schedule was consistent for all patients. However, in a retrospective study , the dose scheduled was never consistent and often adjusted according to conditions. That is exactly why there were different dose densities. From my point of view, such statement can not clear the concerns.
3# Low RDI defined as 15%. What is the exact dose of dexamethasone in this study? This is important is guiding clinical treatment but the true dose is missing. Was reduction pre-planned for treatment or modified during treatment?
Author Response
1# Regardless of analysis method, NDMM and RRMM are different polulations and should not be pooled together for such a study.
Thank you for your important suggestion. We agree with your point suggesting analyzing NDMM and RRMM separately.
When the analyses between NDMM and RRMM were divided, the high RDI of DARA and low RDI of DEX were not associated with long OS and TTNT because the number of events for OS and TTNT was quite low (4 and 5, respectively). In the NDMM group, the RDI of DARA, DEX, and LEN were not associated with the OS (P = 0.173, 0.327, and 0.211, respectively). The 3-year TTNT in the high RDI of LEN group was significantly higher than that in the low RDI of LEN group (94.4% vs. 75.7%, P = 0.035), whereas the RDI of DARA and DEX were not related to TTNT (P = 0.190 and 0.945, respectively). However, using the multivariate analysis for prognostic factors, HRCA and age75 years, in the univariate analysis, the low RDI of LEN was not a significant prognostic factor for TTNT (HR 0.394; 95%CI, 0.044 – 3.550; P =0.406).
Among the RRMM patients, the 3-year OS in the high RDI of DARA and low RDI of DEX groups was significantly higher than that in the low RDI of DARA and high RDI of DEX groups, respectively (88.5% vs 59.4%, P = 0.032, and 83.5% vs 48.6%, P = 0.035). Multivariate analysis showed that a low RDI of DEX was associated with a long OS (HR, 0.276; 95%CI, 0.107 – 0.710, P = 0.008), whereas a high RDI of DARA was not related to OS (P = 0.881). Regarding TTNT in RRMM, the 3-year TTNT in the high RDI of DARA group was significantly longer than that in the low RDI of DARA group (68.8% vs 41.1%, P=0.014) while there was no significant correlation between the RDI of DEX and TTNT (P= 0.240). Using the multivariate analysis, the RDI of DARA tended to be associated with TTNT although there was no statistical significance (HR, 0.509; 95%CI, 0.240 – 1.080; P = 0.079). The RDI of LEN was not associated with OS or TTNT (P = 0.663 and 0.442, respectively).
We added the results in the NDMM and RRMM groups in the results as below.
Line 205 – 225.
When the analyses between NDMM and RRMM were divided, the high RDI of DARA and low RDI of DEX were not associated with long OS and TTNT because the number of events for OS and TTNT was quite low (4 and 5, respectively). In the NDMM group, the RDI of DARA, DEX, and LEN were not associated with the OS (P = 0.173, 0.327, and 0.211, respectively). The 3-year TTNT in the high RDI of LEN group was significantly higher than that in the low RDI of LEN group (94.4% vs. 75.7%, P = 0.035), whereas the RDI of DARA and DEX were not related to TTNT (P = 0.190 and 0.945, respectively). However, using the multivariate analysis for prognostic factors, HRCA and age75 years, in the univariate analysis, the low RDI of LEN was not a significant prognostic factor for TTNT (HR 0.394; 95%CI, 0.044 – 3.550; P =0.406).
Among the RRMM patients, the 3-year OS in the high RDI of DARA and low RDI of DEX groups was significantly higher than that in the low RDI of DARA and high RDI of DEX groups, respectively (88.5% vs 59.4%, P = 0.032, and 83.5% vs 48.6%, P = 0.035). Multivariate analysis showed that a low RDI of DEX was associated with a long OS (HR, 0.276; 95%CI, 0.107 – 0.710, P = 0.008), whereas a high RDI of DARA was not related to OS (P = 0.881). Regarding TTNT in RRMM, the 3-year TTNT in the high RDI of DARA group was significantly longer than that in the low RDI of DARA group (68.8% vs 41.1%, P=0.014) while there was no significant correlation between the RDI of DEX and TTNT (P= 0.240). Using the multivariate analysis, the RDI of DARA tended to be associated with TTNT although there was no statistical significance (HR, 0.509; 95%CI, 0.240 – 1.080; P = 0.079). The RDI of LEN was not associated with OS or TTNT (P = 0.663 and 0.442, respectively).
2# The authors stated the DRd schedule was consistent for all patients. However, in a retrospective study , the dose scheduled was never consistent and often adjusted according to conditions. That is exactly why there were different dose densities. From my point of view, such statement can not clear the concerns.
Thank you for your important suggestion. We agree with your point, and have consequently added information regarding the timing of reducing the dose of LEN and DEX.
We reduced lenalidomide because of adverse events, mainly infection, neutropenia, and skin rash, but did not reduce lenalidomide intentionally or in a response-adopted manner. The lenalidomide dose was not reduced in 47 patients. The median initial dose of lenalidomide was 10 mg; the initial dose of lenalidomide was 25 mg in five patients, 15 mg in six, 10 mg in 22, 5 mg in 13, and 5 mg in one, with dosing every alternate day. The median number of cycles when lenalidomide was reduced for the first time was 2 (range, 1 – 31). Among patients in whom the initial dose of lenalidomide was reduced, 26 did not reduce additionally. The median second dose of lenalidomide was 10 mg, and the second doses of lenalidomide were 15 mg in seven patients, 10 mg in eight, 5 mg in eight, and 5 mg in three, again with dosing every alternate day. The median number of cycles when lenalidomide was reduced for the second time was 6 (range, 2 – 32). Finally, lenalidomide was discontinued lenalidomide because of adverse events.
We intentionally reduced dexamethasone dose, not owing to adverse events, in a response-adopted manner. After the 3rd cycle of DRd, dexamethasone was omitted on the day daratumumab was not administered. Dexamethasone was not reduced for 19 patients. The median initial dose of dexamethasone was 20 mg/day; the initial doses of dexamethasone were 40 mg in two patients, 20 mg in 13, and 8 mg in 4. The median number of cycles when dexamethasone was reduced for the first time was 1 (range: 1 – 14). In patients who received a reduced initial dose of dexamethasone, 52 did not receive a reduced dose of dexamethasone. The median second dose of dexamethasone was 10 mg/day; the second doses were 25, 1, 12 mg and 18, 8, and 4 mg, respectively. The median number of cycles when dexamethasone was reduced for the second time was 4 (range, 1 – 13). Finally, one patient intentionally discontinued dexamethasone because of worsening diabetes.
We added the concrete the timing for reduction of Len and Dex in the results as below.
Line 90 – 91.
We did not reduce lenalidomide intentionally or in a response-adopted manner as well.
Line 131 – 152.
Forty-seven patients did not reduce LEN. Median initial dose of LEN was 10mg for them; the initial dose of LEN was 5 of 25mg, 6 of 15mg, 22 of 10mg, 13 of 5mg, and 1 of 5mg every other day. Median number of cycles when LEN was reduced at the first time was 2 (range 1 – 31). In the patients reducing initial dose of LEN, 26 patients did not reduce LEN additionally. Median second dose of LEN was 10mg for them; the second dose of LEN was 7 of 15mg, 8 of 10mg, 8 of 5mg, 3 of 5mg every other day. Median number of cycles when LEN was reduced at the second time was 6 (range 2 – 32). Finally, fifteen patients discontinued LEN due to adverse events.
Regarding dose of DEX, the median cumulative dose and number of DEX sessions during the overall DRd interval were 236 mg (range, 40 – 1228 mg) and 22 mg (range, 1 – 74 mg), respectively. Dexamethasone was not reduced for 19 patients. The median initial dose of dexamethasone was 20 mg/day; the initial doses of dexamethasone were 40 mg in two patients, 20 mg in 13, and 8 mg in 4. The median number of cycles when dexamethasone was reduced for the first time was 1 (range: 1 – 14). In patients who received a reduced initial dose of dexamethasone, 52 did not receive a reduced dose of dexamethasone. The median second dose of dexamethasone was 10 mg/day; the second doses were 25, 1, 12 mg and 18, 8, and 4 mg, respectively. The median number of cycles when dexamethasone was reduced for the second time was 4 (range, 1 – 13). Finally, one patient intentionally discontinued dexamethasone because of worsening diabetes.
3# Low RDI defined as 15%. What is the exact dose of dexamethasone in this study? This is important is guiding clinical treatment but the true dose is missing. Was reduction pre-planned for treatment or modified during treatment?
Thank you for the suggestion. We added the cumulative dose and duration of DEX administration. The median cumulative dose and number of DEX sessions during the overall DRd interval were 236 mg (range, 40 – 1228 mg) and 22 mg (range, 1 – 74 mg), respectively. We usually reduce the dose of DEX not because of adverse events or in the treatment response adopted manner, but in a pre-planned manner. However, the specific schedule of DEX administration varied because this study was retrospective. For instance, just after DRd was available, the dose of DEX was not changed intentionally but was maintained without reduction according to the original protocol.
We added the cumulative dose and time of DEX during overall DRd interval in the Results as below.
Line 142 – 144.
Regarding dose of DEX, the median cumulative dose and number of DEX sessions during the overall DRd interval were 236 mg (range, 40 – 1228 mg) and 22 mg (range, 1 – 74 mg), respectively.
Round 3
Reviewer 2 Report
Comments and Suggestions for Authors
I have no further comments other than those previously shared.
Reviewer 3 Report
Comments and Suggestions for Authors
More analysis done and this study is better now.